# Submultiplicative Glivenko-Cantelli and Uniform Convergence of Revenues

**Noga Alon**
Tel Aviv University, Israel
and Microsoft Research
`nogaa@tau.ac.il`

**Moshe Babaioff**
Microsoft Research
moshe@microsoft.com

**Yannai A. Gonczarowski**
The Hebrew University of Jerusalem, Israel
and Microsoft Research
`yannai@gonch.name`

**Yishay Mansour**
Tel Aviv University, Israel
and Google Research, Israel
`mansour@tau.ac.il`

**Shay Moran**
Institute for Advanced Study, Princeton
`shaymoran1@gmail.com`

**Amir Yehudayoff**
Technion — IIT, Israel
`amir.yehudayoff@gmail.com`

## Abstract

In this work we derive a variant of the classic Glivenko-Cantelli Theorem, which asserts uniform convergence of the empirical Cumulative Distribution Function (CDF) to the CDF of the underlying distribution. Our variant allows for tighter convergence bounds for extreme values of the CDF.

We apply our bound in the context of *revenue learning*, which is a well-studied problem in economics and algorithmic game theory. We derive sample-complexity bounds on the uniform convergence rate of the empirical revenues to the true revenues, assuming a bound on the $k$th moment of the valuations, for any (possibly fractional) $k > 1$.

For uniform convergence in the limit, we give a complete characterization and a zero-one law: if the first moment of the valuations is finite, then uniform convergence almost surely occurs; conversely, if the first moment is infinite, then uniform convergence almost never occurs.

## 1 Introduction

A basic task in machine learning is to learn an unknown distribution $\mu$, given access to samples from it. A natural and widely studied criterion for learning a distribution is approximating its Cumulative Distribution Function (CDF). The seminal Glivenko-Cantelli Theorem [13, 6] addresses this question when the distribution $\mu$ is over the real numbers. It determines the behavior of the empirical distribution function as the number of samples grows: let $X_1, X_2, \ldots$ be a sequence of i.i.d. random variables drawn from a distribution $\mu$ on $\mathbb{R}$ with Cumulative Distribution Function (CDF) $F$, and let $x_1, x_2, \ldots$ be their realizations. The *empirical distribution* $\mu_n$ is

$$\mu_n \triangleq \frac{1}{n} \sum_{i=1}^{n} \delta_{x_i},$$

where $\delta_{x_i}$ is the constant distribution supported on $x_i$. Let $F_n$ denote the CDF of $\mu_n$, i.e., $F_n(t) \triangleq \frac{1}{n} \cdot \left|\{1 \leq i \leq n : x_i \leq t\}\right|$. The Glivenko-Cantelli Theorem formalizes the statement that $\mu_n$ converges to $\mu$ as $n$ grows, by establishing that $F_n(t)$ converges to $F(t)$, uniformly over all $t \in \mathbb{R}$:

**Theorem 1.1** (Glivenko-Cantelli Theorem, [13, 6]). *Almost surely,*

$$\lim_{n \to \infty} \sup_t \left|F_n(t) - F(t)\right| = 0.$$

Some twenty years after Glivenko [13] and Cantelli [6] discovered this theorem, Dvoretzky, Kiefer, and Wolfowitz (DKW) [12] strengthened this result by giving an almost[1] tight quantitative bound on the convergence rate. In 1990, Massart [17] proved a tight inequality, confirming a conjecture due to Birnbaum and McCarty [3]:

**Theorem 1.2** ([17]). $\Pr\left[\sup_t \left|F_n(t) - F(t)\right| > \epsilon\right] \leq 2\exp(-2n\epsilon^2)$ *for all* $\epsilon > 0$, $n \in \mathbb{N}$.

The above theorems show that, with high probability, $F$ and $F_n$ are close up to some *additive* error. We would have liked to prove a stronger, *multiplicative* bound on the error:

$$\forall t : \left|F(t) - F_n(t)\right| \leq \epsilon \cdot F(t).$$

However, for some distributions, the above event has probability 0, no matter how large $n$ is. For example, assume that $\mu$ satisfies $F(t) > 0$ for all $t$. Since the empirical measure $\mu_n$ has finite support, there is $t$ with $F_n(t) = 0$; for such a value of $t$, such a multiplicative approximation fails to hold.

So, the above multiplicative requirement is too strong to hold in general. A natural compromise is to consider a *submultiplicative* bound:

$$\forall t : \left|F(t) - F_n(t)\right| \leq \epsilon \cdot F(t)^\alpha,$$

where $0 \leq \alpha < 1$. When $\alpha = 0$, this is the additive bound studied in the context of the Glivenko-Cantelli Theorem. When $\alpha = 1$, this is the unattainable multiplicative bound. Our first main result shows that the case of $\alpha < 1$ is attainable:

**Theorem 1.3** (Submultiplicative Glivenko-Cantelli Theorem). *Let* $\epsilon > 0$, $\delta > 0$ *and* $0 \leq \alpha < 1$. *There exists* $n_0(\epsilon, \delta, \alpha)$ *such that for all* $n > n_0$, *with probability* $1 - \delta$:

$$\forall t : \left|F(t) - F_n(t)\right| \leq \epsilon \cdot F(t)^\alpha.$$

It is worth pointing out a central difference between Theorem 1.3 and other generalizations of the Glivenko-Cantelli Theorem: for example, the seminal work of Vapnik and Chervonenkis [24] shows that for every class of events $\mathcal{F}$ of VC dimension $d$, there is $n_0 = n_0(\epsilon, \delta, d)$ such that for every $n \geq n_0$, with probability $1 - \delta$ it holds that $\forall A \in \mathcal{F} : \left|p(A) - p_n(A)\right| \leq \epsilon$. This yields Glivenko-Cantelli by plugging $\mathcal{F} = \{(-\infty, t] : t \in \mathbb{R}\}$, which has VC dimension 1. In contrast, the submultiplicative bound from Theorem 1.3 does not even extend to the VC dimension 1 class $\mathcal{F} = \{\{t\} : t \in \mathbb{R}\}$. Indeed, pick any distribution $p$ over $\mathbb{R}$ such that $p(\{t\}) = 0$ for every $t$, and observe that for every sample $x_1, \ldots, x_n$, it holds that $p_n(\{x_i\}) \geq 1/n$, however $p(\{x_i\}) = 0$, and therefore, as long as $\alpha > 0$, it is never the case that $\left|p(\{x_i\}) - p_n(\{x_i\})\right| \leq p(\{x_i\})^\alpha$.

Our second main result gives an explicit upper bound on $n_0(\epsilon, \delta, \alpha)$:

**Theorem 1.4** (Submultiplicative Glivenko-Cantelli Bound). *Let* $\epsilon, \delta \leq 1/4$, *and* $\alpha < 1$. *Then*

$$n_0(\epsilon, \delta, \alpha) \leq \max\left\{\frac{\ln(6/\delta)}{2\epsilon^2}\left(\frac{\epsilon\delta}{3}\right)^{-\frac{4\alpha}{1-\alpha}}, \quad (D+1)\left(10 \cdot \ln\left(12 \cdot \frac{D+4}{\delta(1-\alpha)}\right)\right)^{\frac{4\alpha}{1-\alpha}}\right\},$$

*where* $D = \frac{\ln(6/\delta)}{2\epsilon^2}\left(\frac{\epsilon\delta}{6} \cdot \ln\left(\frac{1+\alpha}{2\alpha}\right)\right)^{-\frac{4\alpha}{1-\alpha}}$.

Note that for fixed $\epsilon, \delta$, when $\alpha \to 0$ the above bound approaches the familiar $O\left(\frac{\ln(1/\delta)}{\epsilon^2}\right)$ bound by DKW [12] and Massart [17] for $\alpha = 0$. On the other hand, when $\alpha \to 1$ the above bound tends

to $\infty$, reflecting the fact that the multiplicative variant of Glivenko-Cantelli ($\alpha = 1$) does not hold. Theorems 1.3 and 1.4 are proven in the supplementary material.

Note that the dependency of the above bound on the confidence parameter $\delta$ is polynomial. This contrasts with standard uniform convergence rates, which, due to applications of concentration bounds such as Chernoff/Hoeffding, achieve logarithmic dependencies on $\delta$. These concentration bounds are not applicable in our setting when the CDF values are very small, and we use Markov's inequality instead. The following example shows that a polynomial dependency on $\delta$ is indeed necessary and is not due to a limitation of our proof.

**Example 1.5.** For large $n$, consider $n$ independent samples $x_1, \ldots, x_n$ from the uniform distribution over $[0, 1]$, and set $\alpha = 1/2$ and $\epsilon = 1$. The probability of the event

$$\exists i : x_i \leq 1/n^3$$

is roughly $1/n^2$: indeed, the complementary event has probability $(1-1/n^3)^n \approx \exp(-1/n^2) \approx 1 - 1/n^2$. When this happens, we have: $F_n(1/n^3) \geq 1/n >> 1/n^3 + 1/n^{3/2} = F(1/n^3) + \left[F(1/n^3)\right]^{1/2}$. Note that this happens with probability inverse polynomial in $n$ (roughly $1/n^2$) and not inverse exponential.

**An application to revenue learning.** We demonstrate an application of our Submultiplicative Glivenko-Cantelli Theorem in the context of a widely studied problem in economics and algorithmic game theory: the problem of revenue learning. In the setting of this problem, a seller has to decide which price to post for a good she wishes to sell. Assume that each consumer draws her private valuation for the good from an unknown distribution $\mu$. We envision that a consumer with valuation $v$ will buy the good at any price $p \leq v$, but not at any higher price. This implies that the expected revenue at price $p$ is simply $r(p) \triangleq p \cdot q(p)$, where $q(p) \triangleq \Pr_{V \sim \mu}[V \geq p]$.

In the language of machine learning, this problem can be phrased as follows: the examples domain $Z \triangleq \mathbb{R}^+$ is the set of all valuations $v$. The hypothesis space $H \triangleq \mathbb{R}^+$ is the set of all prices $p$. The revenue (which is a gain, rather than loss) of a price $p$ on a valuation $v$ is the function $p \cdot 1_{\{p \leq v\}}$.

The well-known *revenue maximization* problem is to find a price $p^*$ that maximizes the expected revenue, given a sample of valuations drawn i.i.d. from $\mu$. In this paper, we consider the more demanding *revenue estimation* problem: the problem of well-approximating $r(p)$, simultaneously for all prices $p$, from a given sample of valuations. (This clearly also implies a good estimation of the maximum revenue and of a price that yields it.) More specifically, we address the following question: when do the *empirical revenues*, $r_n(p) \triangleq p \cdot q_n(p)$, where $q_n(p) \triangleq \Pr_{V \sim \mu_n}[V \geq p] = \frac{1}{n} \cdot \left|\{1 \leq i \leq n : x_i \geq t\}\right|$, uniformly converge to the true revenues $r(p)$? More specifically, we would like to show that for some $n_0$, for $n \geq n_0$ we have with probability $1 - \delta$ that

$$\left|r(p) - r_n(p)\right| \leq \epsilon.$$

The revenue estimation problem is a basic instance of the more general problem of uniform convergence of empirical estimates. The main challenge in this instance is that the prices are unbounded (and so are the private valuations that are drawn from the distribution $\mu$).

Unfortunately, there is no (upper) bound on $n_0$ that is only a function of $\epsilon$ and $\delta$. Moreover, even if we add the expectation of valuations, i.e., $\mathbb{E}[V]$ where $V$ is distributed according to $\mu$, still there is no bound on $n_0$ that is a function of only those three parameters (see Section 2.3 for an example). In contrast, when we consider higher moments of the distribution $\mu$, we are able to derive bounds on the value of $n_0$. These bounds are based on our Submultiplicative Glivenko-Cantelli Bound. Specifically, assume that $\mathbb{E}_{V \sim \mu}[V^{1+\theta}] \leq C$ for some $\theta > 0$ and $C \geq 1$. Then, we show that for any $\epsilon, \delta \in (0, 1)$, we have

$$\Pr\left[\exists v : \left|r(v) - r_n(v)\right| > \epsilon\right] \leq \Pr\left[\exists v : \left|q(v) - q_n(v)\right| > \frac{\epsilon}{C^{\frac{1}{1+\theta}}} q(v)^{\frac{1}{1+\theta}}\right].$$

This essentially reduces uniform convergence bounds to our Submultiplicative Glivenko-Cantelli variant. It then follows that there exists $n_0(C, \theta, \epsilon, \delta)$ such that for any $n \geq n_0$, with probability at least $1 - \delta$,

$$\forall v : \left|r_n(v) - r(v)\right| \leq \epsilon.$$

We remark that when $\theta$ is large, our bound yields $n_0 \approx O\left(\frac{\ln(1/\delta)}{\epsilon^2}\right)$, which recovers the standard sample complexity bounds obtainable via DKW [12] and Massart [17].

When $\theta \to 0$, our bound diverges to infinity, reflecting the fact (discussed above) that there is no bound on $n_0$ that depends only on $\epsilon, \delta$, and $\mathbb{E}[V]$. Nevertheless, we find that $\mathbb{E}[V]$ qualitatively determines whether uniform convergence occurs in the limit. Namely, we show that

- If $\mathbb{E}_\mu[V] < \infty$, then almost surely $\lim_{n \to \infty} \sup_v |r(v) - r_n(v)| = 0$,
- Conversely, if $\mathbb{E}_\mu[V] = \infty$, then almost never $\lim_{n \to \infty} \sup_v |r(v) - r_n(v)| = 0$.

## 1.1 Related work

**Generalizations of Glivenko-Cantelli.** Various generalizations of the Glivenko-Cantelli Theorem were established. These include uniform convergence bounds for more general classes of functions as well as more general loss functions (for example, [24, 23, 16, 2]). The results that concern unbounded loss functions are most relevant to this work (for example, [9, 8, 23]). We next briefly discuss the relevant results from Cortes et al. [8] in the context of this paper; more specifically, in the context of Theorem 1.3. To ease presentation, set $\alpha$ in this theorem to be $1/2$. Theorem 1.3 analyzes the event where the empirical quantile is bounded by[2]

$$q_n(p) \leq q(p) + \epsilon\sqrt{q(p)},$$
$$q_n(p) \geq q(p) - \epsilon\sqrt{q(p)}.$$

whereas, [8] analyzes the event where it is bounded it by:

$$q_n(p) \leq \tilde{O}\big(q(p) + \sqrt{q(p)/n} + 1/n\big),$$
$$q_n(p) \geq \tilde{\Omega}\big(q(p) - \sqrt{q_n(p)/n} - 1/n\big)$$

Thus, the main difference is the additive $1/n$ term in the bound from [8]. In the context of uniform convergence of revenues, it is crucial to use the upper bound on the empirical quantile as we do, as it guarantees that large prices will not overfit, which is the main challenge in proving uniform convergence in this context. In particular, the upper bound from [8] does not provide any guarantee on the revenues of prices $p >> n$, as for such prices $p \cdot 1/n >> 1$.

It is also worth pointing out that our lower bound on the empirical quantile implies that with high probability the quantile of the maximum sampled point is at least $1/n^2$ (or more generally, at least $1/n^{1/\alpha}$ when $\alpha \neq 1/2$), while the bound from [8] does not imply any non-trivial lower bound.

Another, more qualitative difference is that unlike the bounds in [8] that apply for general VC classes, our bound is tailored for the class of thresholds (corresponding to CDF/quantiles), and does not extend even to other classes of VC dimension 1 (see the discussion after Theorem 1.3).

**Uniform convergence of revenues.** The problem of *revenue maximization* is a central problem in economics and Algorithmic Game Theory (AGT). The seminal work of Myerson [20] shows that given a valuation distribution for a single good, the revenue-maximizing selling mechanism for this good is a posted-price mechanism. In the recent years, there has been a growing interest in the case where the valuation distribution is unknown, but the seller observes samples drawn from it. Most papers in this direction assume that the distribution meets some tail condition that is considered "natural" within the algorithmic game theory community, such as boundedness [18, 21, 19, 1, 14, 10][3], such as a condition known as Myerson-regularity [11, 15, 7, 10], or such as a condition known as monotone hazard rate [15].[4] These papers then go on to derive computation- or sample-complexity

bounds on learning an optimal price (or an optimal selling mechanism from a given class) for a distribution that meets the assumed condition.

A recurring theme in statistical learning theory is that learnability guarantees are derived via a, sometimes implicit, uniform convergence bound. However, this has not been the case in the context of revenue learning. Indeed, while some papers that studied bounded distributions [18, 21, 19, 1] did use uniform convergence bounds as part of their analysis, other papers, in particular those that considered unbounded distributions, had to bypass the usage of uniform convergence by more specialized arguments. This is due to the fact that many unbounded distributions do not satisfy any uniform convergence bound. As a concrete example, the (unbounded, Myerson-regular) *equal revenue distribution*[5] has an infinite expectation and therefore, by our Theorem 2.3, satisfies no uniform convergence, even in the limit. Thus, it turns out that the works that studied the popular class of Myerson-regular distributions [11, 15, 7, 10] indeed could not have hoped to establish learnability via a uniform convergence argument. For instance, the way [11, 7] establish learnability for Myerson-regular distributions is by considering the guarded ERM algorithm (an algorithm that chooses an empirical revenue maximizing price that is smaller than, say, the $\sqrt{n}$th largest sampled price), and proving a uniform convergence bound, not for all prices, but only for prices that are, say, smaller than the $\sqrt{n}$th largest sampled price, and then arguing that larger prices are likely to have a small empirical revenue, compared to the guarded empirical revenue maximizer. This means that the guarded ERM will output a good price, but it does not (and cannot) imply uniform convergence for all prices.

We complement the extensive literature surveyed above in a few ways. The first is generalizing the revenue maximization problem to a revenue estimation problem, where the goal is to uniformly estimate the revenue of all possible prices, when no bound on the possible valuations is given (or even exists). The problem of revenue estimation arises naturally when the seller has additional considerations when pricing her good, such as regulations that limit the price choice, bad publicity if the price is too high (or, conversely, damage to prestige if the price is too low), or willingness to suffer some revenue loss for better market penetration (which may translate to more revenue in the future). In such a case, the seller may wish to estimate the revenue loss due to posting a discounted (or inflated) price.

The second, and most important, contribution to the above literature is that we consider arbitrary distributions rather than very specific and limited classes of distributions (e.g., bounded, Myerson-regular, monotone hazard rate, etc.). Third, we derive finite sample bounds in the case that the expected valuation is bounded for some moment larger than 1. We further derive a zero-one law for uniform convergence in the limit that depends on the finiteness of the first moment. Technically, our bounds are based on an additive error rather than multiplicative ones, which are popular in the AGT community.

## 1.2 Paper organization

The rest of the paper is organized as follows. Section 2 contains the application of our Submultiplicative Glivenko-Cantelli to revenue estimation, and Section 3 contains a discussion and possible directions of future work. The proof of the Submultiplicative Glivenko-Cantelli variant, and some extensions of it, appear in the supplementary material.

## 2 Uniform Convergence of Empirical Revenues

In this section we demonstrate an application of our Submultiplicative Glivenko-Cantelli variant by establishing uniform convergence bounds for a family of unbounded random variables in the context of revenue estimation.

## 2.1 Model

Consider a good $g$ that we wish to post a price for. Let $V$ be a random variable that models the valuation of a random consumer for $g$. Technically, it is assumed that $V$ is a nonnegative random variable, and we denote by $\mu$ its induced distribution over $\mathbb{R}^+$. A consumer who values $g$ at a

valuation $v$ is willing to buy the good at any price $p \leq v$, but not at any higher price. This implies that the realized revenue to the seller from a (posted) price $p$ is the random variable $p \cdot 1_{\{p \leq V\}}$. The *quantile* of a value $v \in \mathbb{R}^+$ is

$$q(v) = q(v; \mu) \triangleq \mu\big(\{x : x \geq v\}\big).$$

This models the fraction of the consumers in the population that are willing to purchase the good if priced at $v$. The expected revenue from a (posted) price $p \in \mathbb{R}^+$ is

$$r(p) = r(p; \mu) \triangleq \mathbb{E}_{\mu}\big[p \cdot 1_{\{p \leq V\}}\big] = p \cdot q(p).$$

Let $V_1, V_2, \ldots$ be a sequence of i.i.d. valuations drawn from $\mu$, and let $v_1, v_2, \ldots$ be their realizations. The *empirical quantile* of a value $v \in \mathbb{R}^+$ is

$$q_n(v) = q(v; \mu_n) \triangleq \tfrac{1}{n} \cdot \big|\{1 \leq i \leq n : v_i \geq v\}\big|.$$

The *empirical revenue* from a price $p \in \mathbb{R}^+$ is

$$r_n(p) = r(p; \mu_n) \triangleq \mathbb{E}_{\mu_n}\big[p \cdot 1_{\{p \leq V\}}\big] = p \cdot q_n(p).$$

The revenue estimation error for a given sample of size $n$ is

$$\epsilon_n \triangleq \sup_p \big|r_n(p) - r(p)\big|.$$

It is worth highlighting the difference between revenue estimation and revenue maximization. Let $p^*$ be a price that maximizes the revenue, i.e., $p^* \in \arg\sup_p r(p)$. The maximum revenue is $r^* = r(p^*)$. The goal in many works in revenue maximization is to find a price $\hat{p}$ such that $r^* - r(\hat{p}) \leq \epsilon$, or alternatively, to bound $r^*/r(\hat{p})$.

Given a revenue-estimation error $\epsilon_n$, one can clearly maximize the revenue within an additive error of $2\epsilon_n$ by simply posting a price $p_n^* \in \arg\max_p r_n(p)$, thereby attaining revenue $r_n^* = r(p_n^*)$. This follows since

$$r_n^* = r(p_n^*) \geq r_n(p_n^*) - \epsilon_n \geq r_n(p^*) - \epsilon_n \geq r(p^*) - 2\epsilon_n = r^* - 2\epsilon_n.$$

Therefore, good revenue estimation implies good revenue maximization.

We note that the converse does not hold. Namely, there are distributions for which revenue maximization is trivial but revenue estimation is impossible. One such case is the *equal revenue distribution*, where all values in the support of $\mu$ have the same expected revenue. For such distributions, the problem of revenue maximization becomes trivial, since any posted price is optimal. However, as follows from Theorem 2.3, since the expected revenue of such distributions is infinite, almost never do the empirical revenues uniformly converge to the true revenues.

## 2.2  Quantitative bounds on the uniform convergence rate

Recall that we are interested in deriving sample bounds that would guarantee uniform convergence for the revenue estimation problem. We will show that given an upper bound on the $k$th moment of $V$ for some $k > 1$, we can derive a finite sample bound. To this end we utilize our Submultiplicative Glivenko-Cantelli Bound (Theorem 1.4).

We also consider the case of $k = 1$, namely that $\mathbb{E}[V]$ is bounded, and show that in this case there is still uniform convergence in the limit, but that there cannot be any guarantees on the convergence rate. Interestingly, it turns out that $\mathbb{E}[V] < \infty$ is not only sufficient but also necessary so that in the limit, the empirical revenues uniformly converge to the true revenues (see Section 2.3).

We begin by showing that bounds on the $k$th moment for $k > 1$ yield explicit bounds on the convergence rate. It is convenient to parametrize by setting $k = 1 + \theta$, where $\theta > 0$.

**Theorem 2.1.** *Let $\mathbb{E}_{V \sim \mu}[V^{1+\theta}] \leq C$ for some $\theta > 0$ and $C \geq 1$, and let $\epsilon, \delta \in (0, 1)$. Set*[6]

$$n_0 = \tilde{O}\left( \frac{\ln(1/\delta)}{\epsilon^2} C^{\frac{2}{1+\theta}} \left( \frac{6 \cdot C^{\frac{1}{1+\theta}}}{\epsilon \delta \ln(1 + \theta/2)} \right)^{4/\theta} \right). \tag{1}$$

*For any $n \geq n_0$, with probability at least $1 - \delta$,*

$$\forall v : \quad \big|r_n(v) - r(v)\big| \leq \epsilon.$$

Note that when $\theta$ is large, this bound approaches the standard $O\left(\frac{\ln(1/\delta)}{\epsilon^2}\right)$ sample complexity bound of the additive Glivenko-Cantelli. For example, if all moments are uniformly bounded, then the convergence is roughly as fast as in standard uniform convergence settings (e.g., VC-dimension based bounds).

The proof of Theorem 2.1 follows from Theorem 1.4 and the next proposition, which reduces bounds on the uniform convergence rate of the empirical revenues to our Submultiplicative Glivenko-Cantelli.

**Proposition 2.2.** *Let* $\mathbb{E}_{V \sim \mu}[V^{1+\theta}] \leq C$ *for some* $\theta > 0$ *and* $C \geq 1$, *and let* $\epsilon, \delta \in (0, 1)$. *Then,*

$$\Pr\left[\exists v : \; \left|r(v) - r_n(v)\right| > \epsilon\right] \leq \Pr\left[\exists v : \; \left|q(v) - q_n(v)\right| > \frac{\epsilon}{C^{\frac{1}{1+\theta}}} q(v)^{\frac{1}{1+\theta}}\right].$$

Thus, to prove Theorem 2.1, we first note that Theorem 1.4 (as well as Theorem 1.3) also holds when $F_n$ and $F$ are respectively replaced in the definition of $n_0$ with $q_n$ and $q$ (indeed, applying Theorem 1.4 to the measure $\mu'$ defined by $\mu'(A) \triangleq \mu(\{-a \mid a \in A\})$ yields the required result with regard to the measure $\mu$). We then plug $\epsilon \leftarrow \frac{\epsilon}{C^{\frac{1}{1+\theta}}}$ and $\alpha \leftarrow \frac{1}{1+\theta}$ into this variant of Theorem 1.4 to yield a bound on the right-hand side of the inequality in Proposition 2.2, whose application concludes the proof.

*Proof of Proposition 2.2.* By Markov's inequality:

$$q(v) = \Pr[V \geq v] = \Pr[V^{1+\theta} \geq v^{1+\theta}] \leq \frac{C}{v^{1+\theta}}. \tag{2}$$

Now,

$$
\begin{aligned}
\Pr\left[\exists v : \; \left|r(v) - r_n(v)\right| > \epsilon\right] &= \Pr\left[\exists v : \; \left|v \cdot q(v) - v \cdot q_n(v)\right| > \epsilon\right] \\
&= \Pr\left[\exists v : \; \left|v \cdot q(v) - v \cdot q_n(v)\right| > \frac{\epsilon}{(v^{1+\theta} \cdot q(v))^{\frac{1}{1+\theta}}} (v^{1+\theta} \cdot q(v))^{\frac{1}{1+\theta}}\right] \\
&\leq \Pr\left[\exists v : \; \left|v \cdot q(v) - v \cdot q_n(v)\right| > \frac{\epsilon}{C^{\frac{1}{1+\theta}}} (v^{1+\theta} \cdot q(v))^{\frac{1}{1+\theta}}\right] \\
&= \Pr\left[\exists v : \; \left|q(v) - q_n(v)\right| > \frac{\epsilon}{C^{\frac{1}{1+\theta}}} q(v)^{\frac{1}{1+\theta}}\right].
\end{aligned}
$$

where the inequality follows from Equation (2). $\qquad\square$

## 2.3 A qualitative characterization of uniform convergence

The sample complexity bounds in Theorem 2.1 are meaningful as long as $\theta > 0$, but deteriorate drastically as $\theta \to 0$. Indeed, as the following example shows, there is no bound on the uniform convergence sample complexity that depends only on the first moment of $V$, i.e., its expectation.

Consider a distribution $\eta_p$ so that with probability $p$ we have $V = 1/p$ and otherwise $V = 0$. Clearly, $\mathbb{E}[V] = 1$. However, we need to sample $m_p = O(1/p)$ valuations to see a single nonzero value. Therefore, there is no bound on the sample size $m_p$ as a function of the expectation, which is simply 1.

We can now consider the higher moments of $\eta_p$. Consider the $k$th moment, for $k = 1 + \theta$ and $\theta > 0$, so $k > 1$. For this moment, we have $A_{p,\theta} = \mathbb{E}[V^{1+\theta}] = p^{\theta/(1+\theta)}$, which implies that $m_p = O\left(1/(A_{p,\theta})^{(1+\theta)/\theta}\right)$. This does allow us to bound $m_p$ as a function of $\theta$ and $\mathbb{E}[V^{1+\theta}]$, but for small $\theta$ we have a huge exponent of approximately $1/\theta$.

While the above examples show that there cannot be a bound on the sample size as a function of the expectation of the value, it turns out that there is a very tight connection between the first moment and uniform convergence:

**Theorem 2.3.** *The following dichotomy holds for a distribution $\mu$ on $\mathbb{R}^+$:*

1. *If $\mathbb{E}_\mu[V] < \infty$, then almost surely $\lim_{n \to \infty} \sup_v \left|r(v) - r_n(v)\right| = 0$.*

2. *If $\mathbb{E}_\mu[V] = \infty$, then almost never $\lim_{n \to \infty} \sup_v \left|r(v) - r_n(v)\right| = 0$.*

*That is, the empirical revenues uniformly converge to the true revenues if and only if $\mathbb{E}_\mu[V] < \infty$.*

We use the following basic fact in the Proof of Theorem 2.3:

**Lemma 2.4.** *Let $X$ be a nonnegative random variable. Then*

$$\sum_{n=1}^\infty \Pr[X \geq n] \leq \mathbb{E}[X] \leq \sum_{n=0}^\infty \Pr[X \geq n].$$

*Proof.* Note that:

$$\sum_{n=1}^\infty 1_{\{X \geq n\}} = \lfloor X \rfloor \leq X \leq \lfloor X \rfloor + 1 = \sum_{n=0}^\infty 1_{\{X \geq n\}}.$$

The lemma follows by taking expectations. $\qquad\square$

*Proof of Theorem 2.3.* We start by proving item 2. Let $\mu$ be a distribution such that $\mathbb{E}_\mu\big[V\big] = \infty$. If $\sup_v v \cdot q(v) = \infty$ then for every realization $v_1, \ldots, v_n$ there is some $v \geq \max\{v_1, \ldots, v_n\}$ such that $v \cdot q(v) \geq 1$, but $v \cdot q_n(v) = 0$. So, we may assume $\sup_v v \cdot q(v) < \infty$. Without loss of generality we may assume that $\sup_v v \cdot q(v) = {}^1\!/_2$ by rescaling the distribution if needed. Consider the sequence of events $E_1, E_2, \ldots$ where $E_n$ denotes the event that $V_n \geq n$. Since $\mathbb{E}_\mu\big[V\big] = \infty$, Lemma 2.4 implies that $\sum_{n=1}^\infty \Pr[E_n] = \infty$. Thus, since these events are independent, the second Borel-Cantelli Lemma [4, 5] implies that almost surely, infinitely many of them occur and so infinitely often

$$V_n \cdot q_n(V_n) \geq 1 \geq V_n \cdot q(V_n) + \tfrac{1}{2}.$$

Therefore, the probability that $v \cdot q_n(v)$ uniformly converge to $v \cdot q(v)$ is 0.

Item 1 follows from the following monotone domination theorem:

**Theorem 2.5.** *Let $\mathcal{F}$ be a family of nonnegative monotone functions, and let $F$ be an upper envelope[7] for $\mathcal{F}$. If $\mathbb{E}_\mu[F] < \infty$, then almost surely:*

$$\lim_{n \to \infty} \sup_{f \in \mathcal{F}} \Big|\mathbb{E}_\mu[f] - \mathbb{E}_{\mu_n}[f]\Big| = 0.$$

Indeed, item 1 follows by plugging $\mathcal{F} = \big\{v \cdot 1_{x \geq v} : v \in \mathbb{R}^+\big\}$, which is uniformly bounded by the identity function $F(x) = x$. Now, by assumption $\mathbb{E}_\mu[F] < \infty$, and therefore, almost surely

$$\lim_{n \to \infty} \sup_{v \in \mathbb{R}^+} \big|r(v) - r_n(v)\big| = \lim_{n \to \infty} \sup_{f \in \mathcal{F}} \Big|\mathbb{E}_\mu[f] - \mathbb{E}_{\mu_n}[f]\Big| = 0. \qquad\square$$

Theorem 2.5 follows by known results in the theory of empirical processes (for example, with some work it can be proved using Theorem 2.4.3 from Vaart and Wellner [22]). For completeness, we give a short and basic proof in the supplementary material.

## 3   Discussion

Our main result is a submultiplicative variant of the Glivenko-Cantelli Theorem, which allows for tighter convergence bounds for extreme values of the CDF. We show that for the revenue learning setting our submultiplicative bound can be used to derive uniform convergence sample complexity bounds, assuming a finite bound on the $k$th moment of the valuations, for any (possibly fractional) $k > 1$. For uniform convergence in the limit, we give a complete characterization, where uniform convergence almost surely occurs if and only if the first moment is finite.

It would be interesting to find other applications of our submultiplicative bound in other settings. A potentially interesting direction is to consider unbounded loss functions (e.g., the squared-loss, or log-loss). Many works circumvent the unboundedness in such cases by ensuring (implicitly) that the losses are bounded, e.g., through restricting the inputs and the hypotheses. Our bound offers a different perspective of addressing this issue. In this paper we consider revenue learning, and replace the boundedness assumption by assuming bounds on higher moments. An interesting challenge is to

prove uniform convergence bounds for other practically interesting settings. One such setting might be estimating the effect of outliers (which correspond to the extreme values of the loss).

In the context of revenue estimation, this work only considers the most naïve estimator, namely of estimating the revenues by the empirical revenues. One can envision other estimators, for example ones which regularize the extreme tail of the sample. Such estimators may have a potential of better guarantees or better convergence bounds. In the context of uniform convergence of selling mechanism revenues, this work only considers the basic class of posted-price mechanisms. While for one good and one valuation distribution, it is always possible to maximize revenue via a selling mechanism of this class, this is not the case in more complex auction environments. While in many more-complex environments, the revenue-maximizing mechanism/auction is still not understood well enough, for environments where it is understood [7, 10, 14] (as well as for simple auction classes that do not necessarily contain a revenue-maximizing auction [19, 1]) it would also be interesting to study relaxations of the restrictive tail or boundedness assumptions currently common in the literature.

### Acknowledgments

The research of Noga Alon is supported in part by an ISF grant and by a GIF grant. Yannai Gonczarowski is supported by the Adams Fellowship Program of the Israel Academy of Sciences and Humanities; his work is supported by ISF grant 1435/14 administered by the Israeli Academy of Sciences and by Israel-USA Bi-national Science Foundation (BSF) grant number 2014389; this project has received funding from the European Research Council (ERC) under the European Union's Horizon 2020 research and innovation programme (grant agreement No 740282). The research of Yishay Mansour was supported in part by The Israeli Centers of Research Excellence (I-CORE) program (Center No. 4/11), by a grant from the Israel Science Foundation, and by a grant from United States-Israel Binational Science Foundation (BSF); the research was done while author was co-affiliated with Microsoft Research. The research of Shay Moran is supported by the National Science Foundations and the Simons Foundations; part of the research was done while author was co-affiliated with Microsoft Research. The research of Amir Yehudayoff is supported by ISF grant 1162/15.

## Footnotes

[1] The inequality due to [12] has a larger constant $C$ in front of the exponent on the right hand side.

[2]For consistency with the canonical statement of the Glivenko-Cantelli theorem, we stated our submultiplicative variants of this theorem with regard to the CDFs $F_n$ and $F$. However, these results also hold when replacing these CDFs with the respective quantiles (tail CDFs) $q_n$ and $q$. See Section 2.2 for details.

[3]The analysis of [1] assumes a bound on the realized revenue (from any possible valuation profile) of any mechanism/auction in the class that they consider. For the class of posted-price mechanisms, this is equivalent to assuming a bound on the support of the valuation distribution. Indeed, for any valuation $v$, pricing at $v$ gives realized revenue $v$ (from the valuation $v$), and so unbounded valuations (together with the ability to post unbounded prices) imply unbounded realized revenues.

[4]Both Myerson-regularity and monotone hazard rate are conditions on the second derivative of the revenue as a function of the quantile of the underlying distribution. In particular, they impose restrictions on the tail of the distribution.

[5]This is a distribution that satisfies the special property that all prices have the same expected revenue.

[6]The $\tilde{O}$ conceals low order terms.

[7]$F$ is an upper envelope for $\mathcal{F}$ if $F(v) \geq f(v)$ for every $v \in V$ and $f \in \mathcal{F}$.

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
