[Supplementary Material]

**Supplementary Material for**
**Submultiplicative Glivenko-Cantelli and Uniform Convergence of Revenues**
**by Alon, Babaioff, Gonczarowski, Mansour, Moran, and Yehudayoff**

## A   Proof of Theorem 1.3

In this appendix $\mu$ is a fixed but otherwise arbitrary distribution, with CDF $F$ and empirical CDF $F_n$.

Theorem 1.3 is a corollary of the following lemma, which gives a quantitative bound on the confidence parameter $\delta$.

**Lemma A.1.** *Let* $n \in \mathbb{N}$, $\epsilon > 0$ *and* $\alpha, p, q \in (0, 1)$. *Assume that* $n \geq \epsilon^{-\frac{1}{1-\alpha}}$ *and* $p \leq \min\{\epsilon^{\frac{1}{1-\alpha}}, 1/e\}$. *Then,*

$$\Pr\Big[\exists t: \ \big|F(t) - F_n(t)\big| > \epsilon \cdot F(t)^\alpha\Big] \leq q + \left\lceil \frac{\ln\ln(\frac{n}{q})}{\ln(\frac{1+\alpha}{2\alpha})} \right\rceil \frac{p^{\frac{1-\alpha}{2}}}{\epsilon} + 2\exp\big(-2n(\epsilon p^\alpha)^2\big).$$

Note that $p$ and $q$ appear only on the right-hand side, and therefore can be "tuned" in order to minimize the upper bound. Our proof of Lemma A.1 uses Theorem 1.2. To better understand the parameters, we state the following corollary (whose first item is stronger than Theorem 1.3).

**Corollary A.2.** *There are constants* $c_1, c_2 > 0$ *so that the following holds.*

1. *If* $\alpha(n) \leq 1 - c_1 \cdot \frac{\ln\ln(n)}{\ln(n)}$, *then the probability of the event*

$$\forall t: \big|F(t) - F_n(t)\big| \leq \epsilon \cdot F(t)^{\alpha(n)}$$

   *tends to* 1 *as* $n$ *tends to* $\infty$.

2. *If* $\alpha(n) \geq 1 - c_2 \cdot \frac{1}{\ln(n)}$ *and* $\mu$ *is uniform over* $[0, 1]$, *then the probability of the event*

$$\forall t: \big|F(t) - F_n(t)\big| \leq \frac{1}{10} \cdot F(t)^{\alpha(n)}$$

   *is at most* $1/2$, *for all* $n \geq 2$.

We leave as an open question to determine the behavior of these probabilities when

$$\alpha(n) \in \left[1 - c_1 \cdot \frac{\ln\ln(n)}{\ln(n)}, 1 - c_2 \cdot \frac{1}{\ln(n)}\right].$$

Corollary A.2 is proven in Appendix D.

*Proof of Lemma A.1.* Let $\epsilon, \alpha, q, p, n$ be as in the statement of the lemma. We partition the event in question to three parts, depending on the value of $t$ as follows. Partition $\mathbb{R}$ to

$$I_{[0,q/n]} = \left\{t \in \mathbb{R}: \ 0 \leq F(t) \leq \frac{q}{n}\right\}, \ I_{(q/n,p)} = \left\{t \in \mathbb{R}: \ \frac{q}{n} < F(t) \leq p\right\}$$

and

$$I_{[p,1]} = \{t \in \mathbb{R}: \ p < F(t) \leq 1\}.$$

There are three corresponding events $E_{[0,q/n]}, E_{(q/n,p)}$ and $E_{[p,1]}$; for example, $E_{[0,q/n]}$ is the event that $\exists t \in I_{[0,q/n]} : B(t) = 1$, where $B(t)$ is the indicator of $\big|F(t) - F_n(t)\big| > \epsilon \cdot F(t)^\alpha$.

The following three claims bound from above the probabilities of these three events. The three claims and the union bound complete the proof of the theorem.

**Claim A.3.** $\Pr\big[E_{[0,q/n]}\big] \leq q$.

*Proof.* Let $t \in I_{[0,q/n]}$ be so that $B(t) = 1$. For any $t \in I_{[0,q/n]}$ we have that $F(t) \leq q/n \leq 1/n \leq \epsilon^{\frac{1}{1-\alpha}}$, where the last inequality is by our assumption on $n$ and $\epsilon$. This implies that $F(t) \leq \epsilon F(t)^\alpha$. Since $B(t) = 1$ it must be the case that $F_n(t) > F(t) + \epsilon\big(F(t)\big)^\alpha \geq 0$, and therefore at least one sample $x_i$ satisfies $x_i \leq t \leq q/n$. Now, by the union bound,

$$\Pr\big[E_{[0,q/n]}\big] \leq \Pr\big[\exists i \in [n]: \ x_i \in I_{[0,q/n]}\big] \leq n \cdot \frac{q}{n} = q. \qquad \square$$

**Claim A.4.** $\Pr\big[E_{(q/n,p)}\big] \leq \left\lceil \frac{\ln\ln(\frac{n}{q})}{\ln(\frac{1+\alpha}{2\alpha})} \right\rceil \frac{p^{\frac{1-\alpha}{2}}}{\epsilon}$.

*Proof.* If $q/n \geq p$ then this event is empty and its probability is $0$. Therefore, assume that $q/n < p$, and that this event is not empty.

For all $t \in I_{(q/n,p)}$ since $F(t) \leq p \leq \epsilon^{\frac{1}{1-\alpha}}$ we have $F(t) - \epsilon F(t)^\alpha \leq 0$. So, it suffices to consider the event

$$\exists t \in I_{(q/n,p)} : F_n(t) > F(t) + \epsilon \cdot F(t)^\alpha.$$

Consider the decreasing sequence of numbers $p_0, p_1, \ldots, p_m$ defined by

$$p_i = p^{\left(\frac{1+\alpha}{2\alpha}\right)^i},$$

where $m$ is such that $p_m < q/n \leq p_{m-1}$. Since $p \leq 1/e$, we can bound $m \leq \left\lceil \frac{\ln\ln(\frac{n}{q})}{\ln(\frac{1+\alpha}{2\alpha})} \right\rceil$. Let

$$t_i = \inf\{t \in I_{(q/n,p)} : F(t) \geq p_i\}.$$

Let $F_n^-(t) = \mu_n\big(\{x : x < t\}\big)$. We claim that

$$\exists t \in I_{(q/n,p)} : B(t) = 1 \implies \exists i < m : F_n^-(t_i) > \epsilon \cdot p_{i+1}^\alpha,$$

Indeed, assume that $t \in I_{(q/n,p)}$ satisfies $F_n(t) > F(t) + \epsilon \cdot F(t)^\alpha$. Since $p_m < t < p_0$, there is some $0 \leq i \leq m-1$ such that $p_{i+1} \leq F(t) < p_i$. Note that $t_{i+1} \leq t < t_i$. Indeed, $t_{i+1} \leq t$ follows since $p_{i+1} \leq F(t)$, and $t < t_i$ follows since $F(t_i) \geq p_i$ (which is implied by right continuity of $F$). Hence,

$$F_n^-(t_i) \geq F_n(t) > F(t) + \epsilon \cdot F(t)^\alpha \geq p_{i+1} + \epsilon \cdot p_{i+1}^\alpha \geq \epsilon \cdot p_{i+1}^\alpha.$$

It hence remains to upper bound the union of these events. Note that

$$\mathbb{E}\big[F_n^-(t_i)\big] = \mu\big(\{x : x < t_i\}\big) \leq p_i.$$

Therefore, by Markov's inequality:

$$\Pr\Big[F_n^-(t_i) > \epsilon \cdot p_{i+1}^\alpha\Big] \leq \frac{p_i}{\epsilon p_{i+1}^\alpha} = \frac{1}{\epsilon} p^{\left(\frac{1-\alpha}{2}\right)\left(\frac{1+\alpha}{2\alpha}\right)^i}.$$

By the union bound,

$$\Pr\Big[\exists i < m : F_n(t_i) > p_{i+1} + \epsilon \cdot p_{i+1}^\alpha\Big] \leq \frac{1}{\epsilon} \sum_{i=0}^{m-1} p^{\left(\frac{1-\alpha}{2}\right)\left(\frac{1+\alpha}{2\alpha}\right)^i} \leq \frac{m}{\epsilon} p^{\frac{1-\alpha}{2}} \leq \left\lceil \frac{\ln\ln(\frac{n}{q})}{\ln(\frac{1+\alpha}{2\alpha})} \right\rceil \frac{p^{\frac{1-\alpha}{2}}}{\epsilon}. \square$$

**Claim A.5.** $\Pr\big[E_{[p,1]}\big] \leq 2\exp\big(-2n(\epsilon p^\alpha)^2\big)$.

*Proof.* For all $t \in I_{[p,1]}$ we have $F(t)^\alpha \geq p^\alpha$. The claim follows by Theorem 1.2. $\square$

Lemma A.1 follows from combining Claims A.3, A.4, and A.5. $\square$

# B  Proof of Theorem 1.4

*Proof.* Let $\epsilon, \delta \leq 1/4$ and $\alpha < 1$. By Lemma A.1,

$$\Pr\Big[\exists t : \big|F(t) - F_n(t)\big| > \epsilon \cdot F(t)^\alpha\Big] \leq q + \left\lceil \frac{\ln\ln(\frac{n}{q})}{\ln(\frac{1+\alpha}{2\alpha})} \right\rceil \frac{p^{\frac{1-\alpha}{2}}}{\epsilon} + 2\exp\big(-2n(\epsilon p^\alpha)^2\big) \quad (3)$$

for every $q \leq 1$, $n \geq \epsilon^{-\frac{1}{1-\alpha}}$ and $p \leq \epsilon^{\frac{1}{1-\alpha}}$.

Set $q, p$ so that each of the first two summands in Equation 3 is at most $\delta$. Specifically, $q = \delta$, and

1. if $\frac{\ln\ln(\frac{n}{\delta})}{\ln(\frac{1+\alpha}{2\alpha})} \geq 1$ then set $p = \left(\epsilon\delta \cdot \frac{\ln\left(\frac{1+\alpha}{2\alpha}\right)}{2\ln\ln(\frac{n}{\delta})}\right)^{\frac{2}{1-\alpha}}$, and

2. if $\frac{\ln \ln(\frac{n}{\delta})}{\ln(\frac{1+\alpha}{2\alpha})} < 1$ then set $p = (\epsilon\delta)^{\frac{2}{1-\alpha}}$.

Note that indeed the requirements $n \geq \epsilon^{-\frac{1}{1-\alpha}}$ and $p \leq \epsilon^{\frac{1}{1-\alpha}}$ are satisfied by $p$ and by the desired $n$ (from the theorem statement).

Plugging these $p$ and $q$ in the third summand in Equation 3 yields:

$$2\exp\left(-2n\epsilon^2\left(\epsilon\delta \cdot \frac{\ln\left(\frac{1+\alpha}{2\alpha}\right)}{2\ln\ln(\frac{n}{\delta})}\right)^{\frac{4\alpha}{1-\alpha}}\right)$$

when $\frac{\ln\ln(\frac{n}{\delta})}{\ln(\frac{1+\alpha}{2\alpha})} \geq 1$, or

$$2\exp\left(-2n\epsilon^2(\epsilon\delta)^{\frac{4\alpha}{1-\alpha}}\right)$$

otherwise. In order for the above to be at most $\delta$, it suffices that

$$2n\epsilon^2\left(\epsilon\delta \cdot \frac{\ln\left(\frac{1+\alpha}{2\alpha}\right)}{2\ln\ln(\frac{n}{\delta})}\right)^{\frac{4\alpha}{1-\alpha}} \geq \ln(2/\delta)$$

when $\frac{\ln\ln(\frac{n}{\delta})}{\ln(\frac{1+\alpha}{2\alpha})} \geq 1$, or

$$2n\epsilon^2(\epsilon\delta)^{\frac{4\alpha}{1-\alpha}} \geq \ln(2/\delta)$$

otherwise. The second case implies an explicit bound of

$$n \geq \frac{\ln(2/\delta)}{2\epsilon^2}(\epsilon\delta)^{-\frac{4\alpha}{1-\alpha}}. \tag{4}$$

To get an explicit bound on $n$ in the first case, we need to solve a recursion of the following type: find a lower bound on $n$ so that the following inequality holds:

$$n \geq D\bigl(\ln\ln(E \cdot n)\bigr)^F,$$

where $D \geq 0$, $E \geq 4$, $F \geq 0$. (Here $D = \frac{\ln(2/\delta)}{2\epsilon^2}\left(\frac{\epsilon\delta}{2} \cdot \ln\left(\frac{1+\alpha}{2\alpha}\right)\right)^{-\frac{4\alpha}{1-\alpha}}$, $E = \frac{1}{\delta}$, $F = \frac{4\alpha}{1-\alpha}$.) Setting

$$n \geq (D+1)\Bigl(10\bigl(\ln(D+4)+\ln(F+4)+\ln(E)\bigr)\Bigr)^F = (D+1)\left(10 \cdot \ln\left(4 \cdot \frac{D+4}{\delta(1-\alpha)}\right)\right)^{\frac{4\alpha}{1-\alpha}} \tag{5}$$

suffices. Therefore, the probability (i.e., the sum of all three summands of Equation 3) is bounded by $3\delta$. Replacing $\delta$ by $\delta/3$ in Equations 4 and 5 (and in the definition of $D$) yields the desired bound on $n_0(\epsilon, \delta, \alpha)$. □

## C  Proof of Theorem 2.5

*Proof.* Let $\epsilon > 0$. Having $\mathbb{E}_\mu[F] < \infty$ implies that there is $v_0 \in \mathbb{R}^+$ such that $\mathbb{E}_\mu\left[1_{\{V \geq v_0\}}F\right] < \epsilon$. Since we can write

$$\mathbb{E}_\mu[f] = \mathbb{E}_\mu\left[f \cdot 1_{\{V \leq v_0\}}\right] + \mathbb{E}_\mu\left[f \cdot 1_{\{V > v_0\}}\right]$$

it suffices to show that almost surely there exist $n_1$ such that

$$(\forall n \geq n_1)\,(\forall f \in \mathcal{F}) : \left|\mathbb{E}_\mu\left[f \cdot 1_{\{V \geq v_0\}}\right] - \mathbb{E}_{\mu_n}\left[f \cdot 1_{\{V \geq v_0\}}\right]\right| \leq 2\epsilon \tag{6}$$

and that almost surely there exist $n_2$ such that

$$(\forall n \geq n_2)\,(\forall f \in \mathcal{F}) : \left|\mathbb{E}_\mu\left[f \cdot 1_{\{V < v_0\}}\right] - \mathbb{E}_{\mu_n}\left[f \cdot 1_{\{V \leq v_0\}}\right]\right| \leq 3\epsilon. \tag{7}$$

We begin by showing Equation (6): the law of large numbers implies that almost surely, there exists $n_1$ such that $\mathbb{E}_{\mu_n}\left[1_{\{V \geq v_0\}}F\right] < 2\epsilon$, for every $n \geq n_1$. Since every $f \in \mathcal{F}$ satisfies $0 \leq f \leq F$, it

follows that $0 \leq \mathbb{E}_\mu\big[1_{\{V \geq v_0\}}f\big] < \epsilon$, and $0 \leq \mathbb{E}_{\mu_n}\big[1_{\{V \geq v_0\}}f\big] < 2\epsilon$ for $n \geq n_1$. This implies Equation (6).

It remains to show Equation (7): set $\epsilon' = \frac{\epsilon}{F(v_0)+1}$. The Glivenko-Cantelli Theorem implies that almost surely there exists $n_2$ such that

$$(\forall n \geq n_2)\,(\forall v \in \mathbb{R}^+): \ \big|q(v) - q_n(v)\big| \leq \epsilon'.$$

Let $f \in \mathcal{F}$. By monotonicity of $f$, it follows that there is a sequence $0 = a_0, a_1, \ldots, a_N = v_0$ such that $f$ does not change by more than $\epsilon$ within each interval $[a_i, a_{i+1})$, (i.e., $\sup_{x,y \in [a_i, a_{i+1})}\big|f(x) - f(y)\big| < \epsilon$). Consider the piecewise constant function

$$f_\epsilon = f(a_0) + \sum_i \big(f(a_{i+1}) - f(a_i)\big)1_{\{V \geq a_i\}}.$$

Note that $f_\epsilon$ gets the value $f(a_i)$ on each interval $[a_i, a_{i+1})$. Thus, $\big|f(v) - f_\epsilon(v)\big| \leq \epsilon$ for every $v \leq v_0$. Therefore, $\big|\mathbb{E}_\mu[f1_{\{V<v_0\}}] - \mathbb{E}_\mu[f_\epsilon 1_{\{V<v_0\}}]\big| \leq \epsilon$ and $\big|\mathbb{E}_{\mu_n}[f] - \mathbb{E}_{\mu_n}[f_\epsilon]\big| \leq \epsilon$. So, it suffices to show that

$$(\forall n \geq n_2): \ \left|\mathbb{E}_\mu\big[f_\epsilon 1_{\{V<v_0\}}\big] - \mathbb{E}_{\mu_n}\big[f_\epsilon 1_{\{V<v_0\}}\big]\right| \leq \epsilon.$$

Indeed, for $n \geq n_2$:

$$
\begin{aligned}
\left|\mathbb{E}_\mu\big[f_\epsilon 1_{\{V<v_0\}}\big] - \mathbb{E}_{\mu_n}\big[f_\epsilon 1_{\{V<v_0\}}\big]\right| &\leq \sum_i \big(f(a_{i+1}) - f(a_i)\big) \cdot \big|q(a_i) - q_n(a_i)\big| \\
&\leq \sum_i \big(f(a_{i+1}) - f(a_i)\big) \cdot \epsilon' && \text{(by definition of } n_2\text{)} \\
&\leq f(v_0) \cdot \epsilon' \\
&\leq \epsilon. && \text{(by definition of } \epsilon'\text{)}
\end{aligned}
$$

$\square$

# D  Proof of Corollary A.2

*Proof.* We begin with the first item. Let $\delta > 0$. It suffices to prove that

$$\Pr\left[\forall t: \big|F(t) - F_n(t)\big| \leq \epsilon \cdot F(t)^{\alpha(n)}\right] \leq 3\delta$$

for a large enough $n$. To this end, we set $q, p$ so that each of the first two summands in Lemma A.1 is at most $\delta$. Specifically,

$$q = \delta$$

and

$$p = \left(\frac{\epsilon\delta \ln\left(\frac{1+\alpha}{2\alpha}\right)}{\ln\ln\left(\frac{n}{\delta}\right) + \ln\left(\frac{1+\alpha}{2\alpha}\right)}\right)^{\frac{2}{1-\alpha}}.$$

As required by the premise of Lemma A.1, $p \leq \epsilon^{\frac{1}{1-\alpha}}$. (The other requirement, $n \geq \epsilon^{-\frac{1}{1-\alpha}}$, will be verified at the end of the proof.)

Plugging these values for $p, q$, the last summand becomes

$$2\exp\big(-2n(\epsilon p^\alpha)^2\big) = 2\exp\left(-2n\epsilon^2\left(\frac{\epsilon\delta \ln\left(\frac{1+\alpha}{2\alpha}\right)}{\ln\ln\left(\frac{n}{\delta}\right) + \ln\left(\frac{1+\alpha}{2\alpha}\right)}\right)^{\frac{4\alpha}{1-\alpha}}\right).$$

We need to verify that the above expression becomes less than $\delta$ for large $n$. Equivalently, that

$$\lim_{n\to\infty} n\left(\frac{\epsilon\delta \ln\left(\frac{1+\alpha}{2\alpha}\right)}{\ln\ln\left(\frac{n}{\delta}\right) + \ln\left(\frac{1+\alpha}{2\alpha}\right)}\right)^{\frac{4\alpha}{1-\alpha}} = \infty.$$

Rewriting $\alpha = 1 - \beta$ gives

$$\lim_{n\to\infty} n \left( \frac{\epsilon\delta \ln\left(1 + \frac{\beta}{2-2\beta}\right)}{\ln\ln\left(\frac{n}{\delta}\right) + \ln\left(1 + \frac{\beta}{2-2\beta}\right)} \right)^{\frac{4-4\beta}{\beta}} = \infty.$$

Since we are focusing on small value of $\beta$ we can assume that $\beta \geq 1/2$. Using that $x/2 \leq \ln(1+x) \leq x$ for $x \in [0,1]$ and $\beta \leq 1/2$, it suffices that we show

$$\lim_{n\to\infty} n \left( \frac{\epsilon\delta\beta/2}{\ln\ln(n/\delta) + 1} \right)^{\frac{1}{\beta}} = \infty,$$

or, by taking "ln", that

$$\lim_{n\to\infty} \left( \ln n - \frac{1}{\beta}\left( \ln(1/\epsilon) + \ln(1/\delta) + \ln(4/\beta) + \ln\left(\ln\ln(n/\delta) + 1\right) \right) \right) = \infty.$$

To this end, it suffices that $1/\beta \ln(1/\beta) \leq \ln(n)/2$, which holds for $\beta \geq c \cdot \ln\ln(n)/\ln(n)$, where $c$ is a sufficiently large constant.

It remains to check that the condition stated in Lemma A.1, that $n \geq \epsilon^{-\frac{1}{1-\alpha}} = \epsilon^{-\frac{1}{\beta}}$, is satisfied. Indeed, for a sufficiently large $n$

$$\epsilon^{-\frac{1}{\beta}} \leq 1/\epsilon^{c \cdot \ln(n)/\ln\ln(n)} = \exp\left(c \ln(1/\epsilon) \cdot \ln(n)/\ln\ln(n)\right) < \exp\left(\ln(n)\right) = n.$$

For the second item, let $Y_1 \leq Y_2 \leq \cdots \leq Y_n$ denote the sequence obtained by sorting $X_1, X_2, \ldots, X_n$. Note that it suffices to show that the probability that

$$Y_1 < \frac{1}{2n}$$

is at least $1/2$: indeed, this event implies that

$$\begin{aligned}
F_n\left(\frac{1}{2n}\right) - F\left(\frac{1}{2n}\right) &\geq \frac{1}{n} - \frac{1}{2n} \\
&= \frac{1}{2n} \\
&\geq \frac{1}{10} \cdot \left(\frac{1}{2n}\right)^{1 - \frac{1}{2\ln n}} &\text{(since } n \geq 2\text{)} \\
&= \frac{1}{10} \cdot F\left(\frac{1}{2n}\right)^{1 - \frac{1}{2\ln n}},
\end{aligned}$$

which implies the conclusion with $c_2 = 1/2$.

Thus, it remains to show that with probability of at least $\frac{1}{2}$, we have $Y_1 < \frac{1}{2n}$:

$$\Pr\left[Y_1 \geq \frac{1}{2n}\right] = \Pr\left[\forall i \leq n : X_i \geq \frac{1}{2n}\right] = \left(1 - \frac{1}{2n}\right)^n \geq \frac{1}{2}. \qquad \square$$