[Reviews · NeurIPS 2017]

Reviewer 1



The paper presents the first Glivenko Cantelli type submultiplicative bound and uses it to derive learning guarantees for the problem of revenue estimation. The paper is well written and the results are certainly not trivial my only reservation with this paper is the fact that the dependence on the confidence parameter \delta is not logarithmic. Normally other relative deviation type bounds achieve a logarithmic rate on the confidence parameter why not in this case? i.e. what makes this problem inherently harder? I have read the rebuttal from the authors and still recommend for acceptance.

Reviewer 2



This paper provides a submultiplicative form of the Glivenk-Cantelli via an appropriately generalized Dvoretzky-Kiefer-Wolfowitz type inequality. Nearly matching lower bounds are given, as well as a zero-one law in terms of the finiteness of the first moment. The results appear to be correct and technically non-trivial. The paper will be of broad interest to the NIPS community.

Reviewer 3



This paper provides a generalization of the Glivenko-Cantelli theorem, a "submultiplicative" compromise between additive and multiplicative errors. The two main results go hand in hand, the first dealing with existence of a sufficient index which guarantees submultiplicativity and the second providing an upper bound on such an index in order to provide ease in applying the results. It is clear that only submultiplicative results are possible due to a simple counterexample given. The proofs of the main results are technical, but mathematically clear. Throughout the paper, the author(s) familiarity with previous work with generalizing the Glivenko-Cantelli theorem is verified, and the novelty of the work is demonstrated. The author(s) apply these results to the problem of estimating revenue in an auction via empirical revenues. It is clear that understanding revenue estimation better will lend itself to revenue maximization. Moreover, investigating the connection between finite moments and uniform convergence in this example can offer a framework for exploring other estimators in a similar way. While the paper is well-written, and I believe the results to be significant and even fundamental, I felt that the example given, while interesting, was quite specialized and detracted somewhat from generality of the paper. Even more detail in the discussion section on the potential impact to the learning theory community would be great. Additional notes: The definition of F_n(t) on lines 22 and 23 is hard to read. The phrasing on lines 66-68 is awkward. Between lines 104 and 105, why not continue with notation similar to that introduced in Theorem 1.3 for consistency?